# Global Molecular Response of *Paracoccidioides brasiliensis* to Zinc Deprivation: Analyses at Transcript, Protein and MicroRNA Levels

**DOI:** 10.3390/jof9030281

**Published:** 2023-02-21

**Authors:** Lorena Alves Mesquita, Alexandre Melo Bailão, Juliana Santana de Curcio, Kassyo Lobato Potenciano da Silva, Gabriel da Rocha Fernandes, Mirelle Garcia Silva-Bailão, Evandro Novaes, Célia Maria de Almeida Soares

**Affiliations:** 1Laboratório de Biologia Molecular, Instituto de Ciências Biológicas, Universidade Federal de Goiás, Goiânia 74690-900, GO, Brazil; 2Informática de Biossistemas & Genômica, Fiocruz Minas, Belo Horizonte 30190-009, MG, Brazil; 3Setor de Genética, Departamento de Biologia, Universidade Federal de Lavras, Lavras 37203-202, MG, Brazil

**Keywords:** zinc, miRNA, proteomics, transcriptomics

## Abstract

Zinc is one of the main micronutrients for all organisms. One of the defense mechanisms used by the host includes the sequestration of metals used in fungal metabolism, such as iron and zinc. There are several mechanisms that maintain the balance in the intracellular zinc supply. MicroRNAs are effector molecules of responses between the pathogen and host, favoring or preventing infection in many microorganisms. Fungi of the *Paracoccidioides* genus are thermodimorphic and the etiological agents of paracoccidioidomycosis (PCM). In the current pandemic scenario world mycosis studies continue to be highly important since a significant number of patients with COVID-19 developed systemic mycoses, co-infections that complicated their clinical condition. The objective was to identify transcriptomic and proteomic adaptations in *Paracoccidioides brasiliensis* during zinc deprivation. Nineteen microRNAs were identified, three of which were differentially regulated. Target genes regulated by those microRNAs are elements of zinc homeostasis such as ZRT1, ZRT3 and COT1 transporters. Transcription factors that have zinc in their structure are also targets of those miRNAs. Transcriptional and proteomic data suggest that *P. brasiliensis* undergoes metabolic remodeling to survive zinc deprivation and that miRNAs may be part of the regulatory process.

## 1. Introduction

Zinc is the second most abundant trace metal in vertebrates and, in eukaryotes, is a cofactor of around 9% of cellular proteins. The metal, denoted by its elemental symbol, without reference to its ionization state, is one of the main micronutrients for all organisms due to its catalytic and structural function in proteins [1]. This metal is known especially for its structural role in the regulation of gene transcription by the Zn fingers, considered the largest group of transcription regulators [2,3,4]. In this way, organisms must maintain Zn at a homeostatic level in cells, since a wide variety of cellular processes are dependent upon this metal [5]. However, this metal can be toxic in large amounts, as it catalyzes cytotoxic reactions, causing the accumulation of ROS (reactive oxygen species) which can lead to cell damage [1,6,7].

Studies on the host–pathogen interaction have shown that during infection Zn becomes a resource to be shared, generating a conflict between the pathogen and host [8]. An important step in host defense is the sequestration of metals such as zinc and iron [9,10]. Since zinc is essential for fungal growth and proliferation, the pathogenic traits in these microorganisms include the ability to scavenge those micronutrients. There are several mechanisms that maintain the balance in the intracellular zinc supply, including metal binding to metallothioneins and storage in intracellular compartments, in addition to the transport into and out of the cell [11,12]. Fungi uptake zinc via metal transporters of the ZIP family, which transport Zn from the extracellular environment, or from subcellular compartments into the cytoplasm [13]. This family has been characterized in pathogenic fungi such as *Candida albicans*, *Cryptococcus neoformans*, *Cryptococcus gattii* and *Aspergillus fumigatus*, and in the model organism *Saccharomyces cerevisiae* (Reviewed by Soares et al., 2020 [14]). The yeast encodes two plasma membrane Zn transporters, the high-affinity transporter Zrt1 and the low-affinity transporter Zrt2 [15,16]. Additionally, Zrt3 functions as a vacuolar zinc exporter [17]. Zip1 and Zip2 are the zinc importers in *Cryptococcus species*. While Zip1 is the main transporter in *C. neoformans* [18], Zip 1 and Zip2 are functionally redundant in *C. gattii* [19]. *A. fumigatus* presents three main zinc importers which are pH specialists. ZrfA and ZrfB are required for zinc utilization at an acidic pH, whereas ZrfC operates at a neutral and alkaline pH [20]. *C. albicans* encodes two zinc importers, Zrt1 and Zrt2, whose function depends on pH as well. Zrt2 works in acidic environments but also allows growth in alkaline zinc-limited conditions, being considered the main zinc transporter in *C. albicans* [21]. At a neutral–alkaline pH, *C. albicans* secretes a small protein, Pra1, that binds to extracellular zinc and delivers the metal to Zrt1 [22].

In addition to the membrane proteins of the ZIP family, CDF transporters also contribute to the maintenance of the balance of cytoplasmic zinc levels. They act by removing zinc from the cytoplasm to the extracellular environment or into organelles, allowing the maintenance of zinc-dependent processes as those in the secretory pathway [13]. In yeast, Zrc1 and Cot1 transport zinc into the vacuole [23,24] while the Zrg17/Msc2 complex supplies metal to the endoplasmic reticulum [25].

Zinc homeostasis in *S. cerevisiae* is regulated by the transcription factor Zap1, which can detect the amount of available zinc and thus induce the expression of its target genes, including Zrt1 and Zap1 itself, during zinc deprivation [26]. Zap1 also regulates genes responsible for adaptation to zinc scarcity, such as those involved in phospholipid synthesis and in response to oxidative stress, a condition experienced by the cells under zinc limitation [27]. The Zap1 ortholog in *A. fumigatus*, ZafA, also mediates homeostatic and adaptive responses to zinc deprivation. ZrfA, ZrfB and ZrfC zinc transporters are ZafA targets, being required for the maintenance of zinc homeostasis. ZafA regulon also includes genes related to iron uptake and ergosterol and gliotoxin biosynthesis as well as those necessary for the response to oxidative stress [28].

Fungi of the *Paracoccidioides* genus are thermodimorphic, presenting the mycelial form at room temperature (25–27 °C) and the yeast form at human body temperature (36 °C) [29]. They are the etiological agents of paracoccidioidomycosis (PCM), a human systemic mycosis with wide distribution in Latin America. The route of infection occurs through the inhalation of conidia that reach the pulmonary alveoli and differentiate into the parasitic yeast form. The clinical importance of PCM has increased due to its frequency, the severity of its clinical and anatomical forms, in addition to its mortality rate. In the pandemic scenario in which the world has found itself since the beginning of 2020, mycosis studies continue to be important as a significant number of patients with COVID-19 developed systemic mycoses, co-infections that complicated their clinical condition [30,31].

Zinc deprivation is a condition that members of the *Paracoccidioides* genus likely face during interaction with the host, since Zn high-affinity membrane transporters are induced when yeast cells are in contact with human plasma or infecting mouse liver cells [32,33]. Zinc homeostasis in that genus may be regulated by a transcription factor homologous to *S. cerevisiae* Zap1, which putatively regulates two zinc membrane transporters, ZRT1 and ZRT2 [34]. In addition to membrane transporters, *Paracoccidioides* spp. presents genes orthologs to COT1 and ZRT3 [34], acting in the storage and utilization of zinc present in the vacuole in other fungi.

Information on adaptive responses to zinc scarcity in the *Paracoccidioides* genus was initiated with studies on *Paracoccidioides lutzii*. Proteomics data defined that zinc deprivation leads to the upregulation of several proteins that are related to response to stress, cell rescue and virulence. Results in *P. lutzii* indicate that low zinc increases the levels of proteins related to response to oxidative stress and gluconeogenesis [35]. In the proteomic analysis of *P. lutzii* membranes, zinc deprivation decreased the protein glycosylation process in the endoplasmic reticulum. The downregulation of proteins that participate in oxidative phosphorylation and cell wall remodeling was also observed [36].

The host–pathogen interaction has been widely studied, and several molecules have been described as part of this process, including microRNAs (miRNAs), a class of small endogenous non-coding RNAs [37]. Pathogens can influence the miRNA expression profile of hosts. Interaction studies between bone marrow-derived macrophages and *Listeria monocytogenes* have demonstrated that it can promote changes in the microRNA expression profile in macrophages [38]. Hosts can also influence the pathogen’s miRNA production or even export microRNAs that will promote gene silencing in the pathogen [39], like the cotton plant that exports two miRNAs to the pathogenic fungus *Verticillium dahlia*; those miRNAs are responsible for silencing genes essential for virulence, such as Clp-1 or HiC-15 [40].

In silico analysis allowed the identification of microRNAs in the genome of *P. brasiliensis* [41]. In addition, microRNAs identified in cDNA libraries obtained from yeast, mycelium and transition from mycelium to yeast cells in this fungus demonstrated that the yeast parasitic form presents a remodeling of the expression of microRNAs when compared to mycelium and transition forms [41]. In the yeast phase genes involved in chitin and glucan metabolism are potential targets of differentially regulated miRNAs. A study integrating the role of microRNAs and iron deprivation revealed that microRNAs potentially regulate genes involved in oxidative phosphorylation and transcription factors related to iron homeostasis, indicating that miRNAs may be part of the cellular response during iron deprivation [42].

Recent advances in the identification of microRNAs in *P. brasiliensis* and their role in the mycelial, yeast and transition phases as well as during iron deprivation indicate that these molecules may play an important role in the infectious process. Therefore, the characterization of microRNAs produced by *P. brasiliensis* in zinc deprivation opens the door to investigate the role of these molecules in regulating the expression of genes involved in the uptake and homeostasis of this micronutrient, an essential process for the establishment of infection in the host. In this study we integrated proteomic and transcriptional studies to obtain a broad overview of the responses of *P. brasiliensis* to zinc deprivation. Our studies demonstrated that miRNAs target key genes in zinc homeostasis. Among the possible targets of differentially expressed microRNAs are genes that code for transporters responsible for zinc homeostasis, such as ZRT1/2 orthologs, COT1 and ZRT3 and transcription factors that have zinc in their structure.

## 2. Materials and Methods

### 2.1. Microorganism and Growth Conditions

*P. brasiliensis* Pb18 (ATCC 32069), in the yeast stage, was cultivated and maintained in solid BHI (Brain Heart Infusion) medium at 36 °C. For zinc deprivation experiments, yeast cells were incubated in MMcM medium (McVeigh/Morton medium; [43]) supplemented with the zinc chelator DTPA (Diethylenetriaminepentaacetic Acid, Sigma Aldrich^®^, St. Louis, MO, USA) [19]. Additionally, glassware was previously treated with 10% (*v*/*v*) nitric acid for 18 h. For the control, the MMcM medium was supplemented with 0.03 mM ZnSO_4_ [35]. DTPA concentration was determined after checking cell viability following fungus exposure to DTPA concentrations in MMcM: 3.125 µM, 6.25 µM, 12.5 µM, 25 µM, 50 µM, 100 µM and 200 µM. After 24 h the culture absorbance in ELISA plate was taken at 640 nm [44].

### 2.2. RNA Extraction and cDNA Sequencing

Yeast cells, after cultivation for three days at 36 °C in BHI liquid medium, were transferred to an MMcM medium supplemented with 0.03 mM zinc sulphate (control) or 100 µM DTPA (test). RNA was extracted using the Trizol method [42] and evaluated by 1% (*w*/*v*) agarose gel electrophoresis and Nanodrop (Life Technologies, Carlsbad, CA, USA) to check purity (OD260/OD280). The extracted RNA was used to obtain libraries of RNAs and microRNAs. The libraries were constructed with the NEBNext^®^ Multiplex Small RNA Library Prep Set for Illumina (Illumina KIT). Initially, adapters were ligated to the 3′ and 5′ regions of the RNA molecules, then cDNAs were synthesized, amplified with SuperScript II Reverse Transcriptase with 100 mM DTT and 5X First Strand Buffer (Invitrogen, Waltham, MA, USA), purified and the size selected from an agarose gel. The sequencing of the samples was performed by GenOne Biotechnologies (www.GenOne.com.br accessed on 25 February 2019) Rio de Janeiro-RJ-Brazil, using the Illumina HiSeq 2500 platform.

### 2.3. Identification of MicroRNAs

After obtaining RAW sequences, the quality of the data was evaluated with the FastQC tool (https://www.bioinformatics.babraham.ac.uk/projects/fastqc/ (accessed on 12 August 2019)). The removal of adapters and of low-quality sequences was carried out with the Trimmomatic program [45]. After processing, a new quality control was carried out with FastQC, verifying the complete removal of the adapters and, consequently, the reduction in the size of the sequences, which had, for the most part, reduced between 23 to 33 bp.After processing the sequences, they were submitted to BLASTx [46] against the nr database (https://blast.ncbi.nlm.nih.gov/ (accessed on 4 September 2019)) in order to confirm that the sequences of the RNAseq were from *P. brasiliensis*. The genome utilized was the *Paracoccidioides brasiliensis* Pb18 v.2 (assembly Paracocci_br_Pb18_V2) available on NCBI (https://www.ncbi.nlm.nih.gov/genome/334 (accessed on 4 September 2019)). The identification of microRNAs in *P. brasiliensis* was performed by the miRDeep2 program [47]. Initially, the small RNAs were mapped into the genome of *P. brasiliensis* using the mapper.pl script, with the options-r 30-q-l 17. Then, the script miRDepp2.pl was used, with default parameters, to identify regions with a pre-miRNA signature among the locus with several small mapped RNAs. Finally, the expression of genes with a pre-miRNA signature was quantified, with default parameters from the quantifier.pl script [47]. The quantification of the number of reads mapped in each library was used in the analysis of the differential expression between samples obtained from yeast cells in the presence or not of zinc. The small RNA sequences were submitted to the Short Read Archive of the NCBI under BioProject number PRJNA931606 and are available here: https://www.ncbi.nlm.nih.gov/bioproject/PRJNA931606 (accessed on 5 February 2023).

### 2.4. Analysis of the Expression and Prediction of Target Genes of P. brasiliensis MicroRNAs

The RNAhybrid program [48] was employed to predict the possible target genes of miRNAs. Possible miRNA targets were predicted using 3′UTR and 5′UTR sequences from all *P. brasiliensis* genes. The 3′ UTR and 5′ sequences were defined as the first and last 200 bases of each transcript, respectively. The transcript sequences were obtained from the NCBI database (https://www.ncbi.nlm.nih.gov/genome/334?genome_assembly_id=212342 (accessed on 26 December 2019)) and the selection of the last 200 bases was performed with a custom script. The set of *P. brasiliensis* transcripts with a high potential to undergo the silencing process by microRNAs were classified using the FungiFun tool (https://sbi.hki-jena.de/fungifun/ (accessed on 2 February 2020)).

### 2.5. Preparation of Protein Extracts

For the preparation of protein extracts, yeast cells were cultivated in the same conditions cited above to obtain RNAs. The cells were collected by centrifugation, and an extraction buffer containing 20 mM Tris–HCl, 2 mM CaCl_2_, pH 8.8 was added. The suspension was distributed in tubes containing glass spheres and processed in the Bead Beater equipment (BioSpec, OKC, OK, USA) for 5 cycles of 30 s. Lysed cells were centrifuged at 10,000× *g* for 15 min at 4 °C and proteins in the supernatant were quantified by the Bradford method [49].

### 2.6. Proteomic Analysis by Liquid Chromatography Coupled to NanoUPLC-MS^E^ Mass Spectrometry, Data Processing and Protein Identification

A total of 200 µg/µL of two samples (control and test) were prepared for NanoUPLC-MSE as previously described by Murad et al. (2011) [50]. Peptides were separated by Ultra High-Performance Liquid Chromatography according to Baeza et al. (2017) [51], using the system ACQUITY UPLC^®^ M-Class (Waters Corporation, Manchester, UK) coupled to the mass spectrometer Synapt G1 HDMS ™ (Waters Micromass, Manchester, UK). The samples were separated by liquid chromatography by the nanoACQUITY system (Waters Corporation, Manchester, UK), using gradual increasing concentrations (10, 14, 16, 20 and 65%) of acetonitrile (ACN) (Sigma-Aldrich, St. Louis, MO) for the release of peptides. Protein identification and data analysis were performed according to Geromanos et al. (2009) [52]. To determine the differentially expressed proteins, changes of at least ±1.5 were considered. The functional classification of proteins was performed by the Funcat system using the FungiFun tool (https://sbi.hki-jena.de/fungifun/ (accessed on 3 March 2020)) [53]. The determination of zinc-binding regions in the identified proteins was performed using the metal PDB database (https://metalpdb.cerm.unifi.it/ (accessed on 1 April 2020)), which provides knowledge about metallic sites in biological macromolecules, from the structural information contained in the Protein Data Bank [54]. The proteome was deposited in the ProteomeXchange via the PRIDE database with the accession number: PXD039807.

### 2.7. Proteomic Data Analysis

The data consisted of three replicates containing global proteome. Acceptance criteria was applied to increase the reliability, as following: two peptides per protein at minimum; five minimum fragments ions match per protein and one minimum peptide matches per protein. Expression data were normalized as described in de Curcio et al. (2017) [36]. Briefly, to normalize the data, a protein present in all replicates with a coefficient of variance lower than 10% was used and after comparing the expression between the conditions with the presence and absence of zinc. The tolerated mass variation was 50 ppm.

### 2.8. mRNA-Seq Analysis

Aliquots of RNA samples were used for the construction of cDNA libraries. cDNAs were amplified, purified and size selected from an agarose gel and sequencing was performed by the GenOne Biotechnologies company (www.GenOne.com.br) using the Illumina HiSeq 2500 platform. The reference genome of *P. brasiliensis* was used for the annotation of data from the sequencing through the ENSEMBL program, version 44 (https://www.ensembl.org/index.html (accessed on 27 August 2019)) [55]. The reference genome was indexed, and readouts mapped and aligned using STAR software [56]. Readings mapped to more than one locus were deleted [57]. The mRNA sequences were submitted to the NCBI under BioProject number PRJNA931668 and are available here: https://www.ncbi.nlm.nih.gov/bioproject/PRJNA931668 (accessed on 4 February 2023).

### 2.9. Statistical Analysis

For statistical analysis of cell viability, the Student’s *t*-test was used and *p*-values ≤ 0.05 were considered statistically significant. The program edgeR package in R (Robinson et al., 2010) [58] was used for the differential expression analysis of microRNAs, comparing samples obtained from yeast cells deprived or not of zinc. MicroRNAs of *P. brasiliensis* expressed only in zinc deprivation (test) or increased or decreased of at least 2-fold in zinc deprivation, compared to the control, were used to identify possible target genes. The quantitative data analysis from protein extracts was performed according to Geromanos et al. (2009) [52]. To determine the differentially expressed proteins, changes of at least ±1.5 were considered. For mRNA-seq, the differently expressed genes were identified by the edgeR package in R (Robinson et al., 2010) [58].

### 2.10. Construction of Heat Maps and MicroRNAs Images

The heat map of the differentially expressed microRNAs was constructed by heatmap function from package “base” in R 4.0 [59]. The images of the miRNAs were obtained by the RNA fold Platform through the precursor sequences (http://rna.tbi.univie.ac.at/cgibin/RNAWebSuite/RNAfold.cgi (accessed on 2 May 2022)) [41].

## 3. Results

### 3.1. Viability of Cells Exposed to Zinc Deprivation

To evaluate the effect of Zn deprivation on the viability of *P. brasiliensis*, yeast cells were grown in a wide range of the Zn chelator DTPA. The data demonstrated that cell viability is not affected by Zinc deprivation up to 50 µM. At 100 and 200 µM DTPA the viability decreased to 80% of control cells, but the difference was not statistically significant (Appendix A). This suggests *P. brasiliensis* presents molecular mechanisms to adapt to zinc-deprivation imposed by up to 200 µM DTPA. Therefore, based on the results of the analysis described above, 100 µM of DTPA was set for further analyses, since it was the highest DPTA concentration that did not substantially affect fungal viability.

### 3.2. Identification of MiRNAs

In order to identify the miRNA population regulated by Zn availability, micro-RNA libraries from *P. brasiliensis* yeast cells grown under zinc deprivation and control conditions were constructed and subjected to deep sequencing. The approach generated from 21,406,924 to 26,556,055 raw sequences which were subjected to computational processing. The quality analysis of the microRNA-seq analyses showed that most of the generated sequences reached a Phred score above 30, which means an average of one sequencing error for every 1000 base pairs, indicating the confidence of the deep sequencing data (Appendix A). Approximately, 85% of the sequences mapped to the reference genome. After mapping, the microRNA prediction analysis led to the identification of 19 microRNAs in the six libraries used. Table 1 shows the sequences of pre-microRNAs, mature microRNAs and star sequences found in the two conditions. After statistical analysis, three miRNAs were differentially regulated by zinc availability. Two miRNAs were induced in zinc deprivation (PbZn-miR-1 and PbZn-miR-9), and one was repressed during zinc deprivation (PbZn-miR-4), as depicted in Figure 1a. The predicted structures of the three differentially expressed miRNAs are shown in Figure 1b.

### 3.3. Some Prominent RNAs Putatively Regulated by Micro RNAs during Zinc Deprivation

The putative target genes of the regulated miRNAs in zinc deprivation were identified by bioinformatics analysis based on their complementarity to the 3′ and 5′ UTR. The differentially expressed miRNAs PbZn-miR-1, PbZn-miR-9 and PbZn-miR-4 targeted 124, 913 and 260 genes, respectively The identified targets encode for proteins related to many biological processes such as: amino acid, nitrogen, carbohydrate, nucleotide and lipid metabolisms, energy-producing pathways, transcription, translation, protein processing and degradation, transport, cell defense, signaling. As this work focused on the Zinc homeostasis control in Paracoccidioides sp., the subsequent insights were focused on metal/zinc homeostasis genes and zinc/metal-dependent proteins. The induced microRNAs PbZn-miR-1 and PbZn-miR-9, had among their targets putative zinc finger-containing transcription factors (23 targets) and alcohol dehydrogenases (3 targets) (Appendix A). The Zn-dependence of those proteins suggests they were repressed by the induced microRNAs, which could save Zn in the cell. PbZn-miR-9 was not described in other studies involving miRNAs from *P. brasiliensis*; this microRNA targets mRNAs relevant for the transportation of micronutrients such as the mitochondrial iron transporter (PADG_04903), the vacuolar zinc transporters ZRT3 (PADG_05322) and COT1 (PADG_08196) (Appendix A). Targeting ZRT3 is quite surprising since it is expected to transport Zn from vacuole to the cytoplasm in low zinc. The fact that COT1 emerged as a target of PbZn-miR-9 was expected since COT1 pumps Zn into the vacuole in high zinc and, thus, the induction of PbZn-miR-9 would result in a decreased expression of COT in low Zn. The predicted targets of the repressed PbZn-miR-4 were mainly related to transport routes, as well as the transport of compounds and ions (Appendix A). One of the targets of PbZn-miR-4 is the zinc transporter ZRT1 (PADG_08567), which was expected since this transporter is required to be induced in low zinc. Thus, the repression of PbZn-miR-4 may result in the induction of Zrt1 expression. The metal homeostasis protein BSD2 is also a targeted PbZn-miR-4. BSD2 is known to traffic the manganese transporters Smf1p and Smf2p of baker’s yeast to the vacuole for degradation under a metal-replete condition [61,62]. The PbZn-miR-4 may be acting on Bsd2 expression to guarantee the correct levels of Zn. Moreover, this mechanism may represent a new connection between Zn and Mn.

### 3.4. Transcriptional and Proteomic Analysis Support the Regulation by MicroRNAs

To obtain additional insights in the regulation performed by microRNAs during zinc deprivation, putative targets of the Zn-regulated microRNAs were searched against regulated genes and proteins identified by RNAseq and proteomic analyses. Five and eleven targets of the differentially expressed miRNAs were regulated in the transcriptome and proteome data, respectively (Table 2). The regulated targets of the induced PbZn-miR-1 and PbZn-miR-9 act on carbon and amino acid metabolism, energy-producing pathways, alcoholic fermentation and transport. The three regulated targets of the repressed PbZn-miR-4 are related to energy production, translation and cell signaling. No regulated targets related to Zn uptake and distribution were found. It is important to highlight that mRNAseq and microRNA data were complementary and not redundant since a low correlation between the two approaches was found.

### 3.5. Metabolic Changes Indicated by Proteome and Transcriptome Analysis

Transcriptome and proteomic results also provided a global view of the global response of *P. brasiliensis* cells exposed to a poor zinc environment. Decreased zinc availability regulated 216 proteins, when the cut of ≤1.5 fold change was used. From those, 80 proteins were down-regulated (Appendix A) and 136 were up-regulated (Appendix A). To identify the Zn-dependent proteins regulated during zinc starvation, a search in the Metal PDB database was performed (https://metalpdb.cerm.unifi.it/ (accessed on 1 April 2020)). Such an approach identified 15 Zn-binding repressed proteins and 22 Zn-binding upregulated proteins (Appendix A). The downregulation of Zn-dependent proteins can avoid metal consumption by non-essential processes which save Zn for more essential processes in fungal cells.

Proteomic data indicated a metabolism remodeling in fungal cells to survive the zinc deficit. The induction of beta-oxidation, amino acid-degrading and TCA cycle enzymes, as well as proteins of respiratory chain indicates that *P. brasiliensis* is using lipids and amino acids as energy sources in low zinc. The anaplerotic PEP carboxykinase (PADG_08503) enzyme was likely upregulated to provide oxaloacetate to TCA and methylcitrate cycle, increasing their processivity to metabolize the acetyl-coA and carbon skeletons produced by fatty acid oxidation and amino acid degradation. Methyclcitrate cycle was putatively activated since methylcitrate dehydratase (PADG_04718) was increased. Two alcohol dehydrogenases (PADG_11405; PADG_04701) were repressed confirming that the fungus was preferentially using the aerobic oxidation of amino acids and lipids.

The decrease in zinc availability resulted in the accumulation of enzymes related to glucose and chitin metabolism. This fact suggests that the fungal cell remodels the cell wall to cope with the nutrient scarcity. This observation was reinforced by the decrease in the levels of enzymes of the cell wall/surface carbohydrate metabolism dTDP-4-dehydrorhamnose reductase (PADG_02719) and UDP-galactopyranose mutase (PADG_00912). Additionally, three enzymes of the pentose phosphate pathway were induced: 6-phosphogluconate dehydrogenase (PADG_03651), transketolase (PADG_04604) and 6-phosphogluconolactonase (PADG_07771, probably to produce NADPH as a reducing power to fight oxidative stress imposed by poor zinc condition.

Transcriptome analysis resulted in 31,122,144 reads on average for the six samples (triplicate for each condition). mRNA-seq read processing resulted in the mapping of 93.82% reads to the *P. brasiliensis* genome database. The mapped readings were analyzed by the STAR software (http://code.google.com/p/rna-star/ (accessed on 27 August 2019)), and the numbers of readings mapped in each replica are presented in Appendix A. To determine the differentially regulated transcripts, a 1.5-fold cutoff was used. A total of 22 upregulated and 33 repressed transcripts were identified (Appendix A). The decreased zinc transcripts encoded to the Zn-dependent enzymes: two putative alcohol dehydrogenases (PADG_01174 and PADG_03859) and O-methyltransferase (PADG_02839). One transcript encoding the zinc transporter (ZRT2) belonging to the ZIP family was induced (PADG_06417). Although, there are no functional studies on Zn-transporters in *P. brasiliensis*, the induced Zn-transporter likely increased the metal availability in cytoplasm as expected for a ZIP transporter.

### 3.6. Integrative Molecular Mechanisms that Promote Fungal Adaptation to Low Zinc

The integration of microRNA, proteomic and transcriptomic analyses offered a wide view of the molecular mechanisms used by *P. brasiliensis* to adapt to low zinc. The data revealed that adaptive processes are regulated by different mechanisms in an integrative fashion. Moreover, the use of the multiomic approaches identified the homeostatic and adaptive responses to zinc deprivation. *P. brasiliensis* obtained enough zinc by regulating zinc transport increasing PbZn-miR-9 and decreasing PbZn-miR-4 which led to an increase in ZRT1 and a decrease in COT1 expression, respectively. The fungus also upregulated an additional ZRT2 in a microRNA-independent fashion, as revealed by the RNAseq approach. The proteomic analysis provided insights on the adaptive response of the pathogen to zinc scarcity. The activation of lipid oxidation, amino acid catabolism, TCA and oxidative phosphorylation pathways and decrease in alcohol dehydrogenases were observed in low Zn. Additionally, the regulation of sugar metabolism as well as cell wall related enzymes indicated that a nutrient-poor environment demands changes in cell wall structure. Likewise, the pentose phosphate pathway seemed to be necessary to fight ROS accumulation promoted by metal limitation. A suggested model for Zn metabolism regulation is depicted in Figure 2.

## 4. Discussion

In the host–pathogen interaction there is a constant dispute for macro and micronutrients. While hosts have mechanisms to deprive the pathogen of these nutrients, pathogens have homeostatic and adaptive mechanisms to survive these stressful conditions and establish infection [63]. Studies already carried out have indicated that zinc deprivation is directly related to the reduction in the growth and virulence of pathogenic fungi such as *C. gattii* [19]. Fungal growth inhibition by zinc deprivation reinforces metal role as an essential structural and catalytic cofactor for many proteins [64]. To evaluate the sensitivity of the fungus *P. brasiliensis* to limiting zinc conditions, the growth and cell viability of yeast cells in zinc scarcity were evaluated. The results showed that zinc scarcity imposed by up to 200 µM DTPA did not significantly affect fungal viability, as shown in the dimorphic pathogenic fungus *Histoplasma capsulatum* [65]. Thus, *P. brasiliensis* harbors efficient mechanisms to support survival and growth in a Zn-limiting condition.

Even with the mechanisms of zinc sequestration imposed by the host on pathogenic fungi, they still manage to survive and establish the infection. Proteomic studies are commonly used to establish the metabolic changes that occur in fungi for survival in a zinc-deficient environment [35,66]. Previous studies in the *Paracoccidioides* genus have indicated metabolic changes and cell wall remodeling when the fungus is subjected to zinc deprivation [35,36]. Proteomic data revealed a change in energy metabolism with the induction of amino acid and fatty acid oxidations. Changes in energy production were observed in some fungi during zinc deprivation, such as *Aspergillus niger*, *H. capsulatum* and *P. lutzii*. In *A. niger* and *P. lutzii* there was a decrease in glycolysis and induction of enzymes involved in gluconeogenesis [35,67]. In *H. capsulatum* there was also a reduction in glycolysis, with the induction of fatty acid oxidation [65]. With the glycolytic pathway and fermentation repressed, other energy sources are being used in fungi during zinc deprivation, thus keeping the energy metabolism active through beta-oxidation and the pentose pathway.

In this work, alterations in enzymes related to fermentation and amino acid synthesis/degradation were also observed. Fermentation is a process remarkably repressed in fungi when zinc availability is scarce, which can be explained by the fact that alcohol dehydrogenases (Adh) are a group of proteins that require zinc as a cofactor for catalytic activity [68]. There are different alcohol dehydrogenases described in *S. cerevisiae* and most are zinc-dependent [69]. Proteomic data from *H. capsulatum* and *P. lutzii* also demonstrated a decrease in Adh in zinc deprivation, which was confirmed by Adh enzymatic activity for *H. capsulatum* and by transcript level in *P. lutzii* [35,65]. In studies in *S. cerevisiae* during zinc deprivation, not only the enzymes’ alcohol dehydrogenases, but several other proteins with zinc-binding sites or zinc in their structure were repressed, such as transcription factors and ribosomal units [66]. The analysis of the presence of zinc in the structure of proteins is relevant since this metal is part of the structural component or cofactor of several proteins. Furthermore, studies with *S. cerevisiae* have already demonstrated a decrease in the expression of proteins directly linked to zinc under conditions of zinc deprivation [66]. The decrease in some proteins can be explained not only by the direct binding to zinc, as in the case of Adh, but also by the fact that some precursors or transcription factors that are part of the regulation of the protein have zinc in their structure. In this sense, S-adenosylmethionine synthase of *S. cerevisiae* is regulated by two transcription factors that have zinc finger domains, Met31 and Met32 [70].

In the present work, in response to zinc deprivation, Hsp90 was differentially regulated. Previous work has shown that Hsp90 is essential for the cell viability of *P. brasiliensis* [71]. Among the functions of this chaperone, one participates in the folding, stabilization, activation and assembly of several proteins. In cells with a reduced expression of Hsp90, there is a decrease in cell viability in stressful environments such as acidic pH and imposed oxidative stress, indicating the importance of this protein for the response to stress conditions [72]. Due to the structural role of zinc, several proteins are in inadequate conformation in metal deprivation, a fact that also contributes to heat shock proteins (Hsp) being induced, as they play a fundamental role in protein folding and remodeling [73,74]. Another protein induced during zinc deprivation is chitin synthase. It has been described that there is a strengthening of the cell wall when the fungus is subjected to a stressful environment, especially regarding the increase in chitin content because of a higher synthesis [36]. Moreover, zinc deprivation promotes changes in the cell wall structure. *H. capsulatum* cells present an increase in chitin and glucan as a response to low zinc, which results in cell surface smoothing [65]. The putative cell wall remodeling in *P. brasiliensis* was reinforced by the regulation of several enzymes related to carbohydrate and cell wall components, as observed in proteomic, transcriptomic and microRNA approaches.

Although several studies in fungi integrate proteomic and transcriptomic analyses, there is not always a correlation between the two results [75,76]. Changes in transcript abundance are not always detected in the same way by the proteome and contrariwise. Although there are studies in which differences in transcripts correspond to changes in the number of proteins [77], this was not observed in this study. Factors such as the cellular stability of proteins and RNAs can lead to differences between proteomic and transcriptomic results in the same condition [77,78].

ZTR membrane transporters of the Zip family are responsible for the uptake of zinc into the cell. One of these zinc transporters is the target of miRNA that is repressed in zinc deprivation; it is inferred that it may be induced since the cell has to capture zinc to supply the need for this metal. It was possible to observe the induction, at the transcriptional level, of the gene encoding to a putative high-affinity zinc transporter ZRT2 (PADG_06417) during zinc deprivation. The first changes to maintain zinc homeostasis when there is limited availability occur at the transcriptional level [15,16,79]. There is an induction of genes encoding zinc transporters, mainly those with high affinity such as ZRT1p in *S. cerevisiae* [80]. In the *Paracoccidiodes* genus the induction of the high-affinity zinc transporter, in this case ZRT2, was already observed when yeast cells of this fungus were cultivated in medium with zinc chelator [35,81].

MiRNAs and their role in host–parasite interactions have been explored in organisms such as Enteroviruses, in addition to being part of the mammalian response to bacterial infections by *Salmonella enterica* and *Mycobacterium* spp [82,83], as examples. To understand how microRNAs can participate in the adaptation to stress conditions, in this work, libraries of microRNAs obtained during the growth of *P. brasiliensis* in zinc deprivation were built. Micro RNAs were identified and compared to previous descriptions in *P. brasiliensis*. Nine microRNAs were identified in a previous work focusing on microRNAs from *P. brasiliensis* present in different stages of the fungus differentiation [60]; one was present in yeast cells submitted to iron deprivation [42]. The iron regulated microRNAs designated PbFe-miR6, PbFe-miR33 and PbFe-miR42 act on processes such as the response to oxidative stress and oxidative phosphorylation [42]. In the present work, the differentially regulated microRNAs presented as probable targets several elements of zinc homeostasis, such as the membrane transporter Zrt1 and the vacuolar transporters Cot1 and Zrt3. The induction of PbZn-miR-9 probably silences COT1 that is dispensable in a poor Zn scenario, since it pumps the metal into the vacuole in Zn-rich conditions. Moreover, the repression of PbZn-miR-4, which targets ZRT1, likely led to an increased expression of this Zn importer. Thus, zinc homeostasis in *P. brasiliensis* is probably regulated not only by transcription factors but also by microRNAs. Additionally, the transcript encoding to a putative high-affinity zinc transporter, ZRT2, was induced in metal deprivation. As previously reported, the response to zinc deprivation includes the induction of these membrane transporters responsible for the homeostasis of this metal. The induction of high-affinity membrane transporters occurs in several fungi, such as *S. cerevisiae*, *A. fumigatus*, *H. capsulatum* and *P. lutzii* [35,65,84,85].

Several transcription factors are possibly repressed by microRNAs during zinc deprivation as described in this work. Thus, as zinc is scarce, there is a decrease in these transcription factors, since many of them require zinc in their constitution, as is the case of transcription factors that have zinc finger domains. Transcription factors have been described as targets of miRNAs in fungi, such as *Trichophyton rubrum* and *P. brasiliensis* [42,86]. In *P. brasiliensis*, during iron deprivation, the pH-responsive transcriptional factor pacC/Rim101 is regulated by the miRNA PbFe_miR27.

In general, *P. brasiliensis* presented metabolic alterations due to zinc availability through the regulation of metabolic pathways that contribute to the fungus’s adaptation to a limited zinc environment. Furthermore, it was brought for the first time the role that miRNAs play in conditions of zinc deprivation. miRNAs act in zinc homeostasis regulating zinc-dependent proteins, transcription factors and transporters. Thus, *P. brasiliensis* survives in low zinc by regulating homeostatic and adaptive mechanisms by a joint action of microRNAs and transcription factors.

## Figures and Tables

**Figure 1 jof-09-00281-f001:**
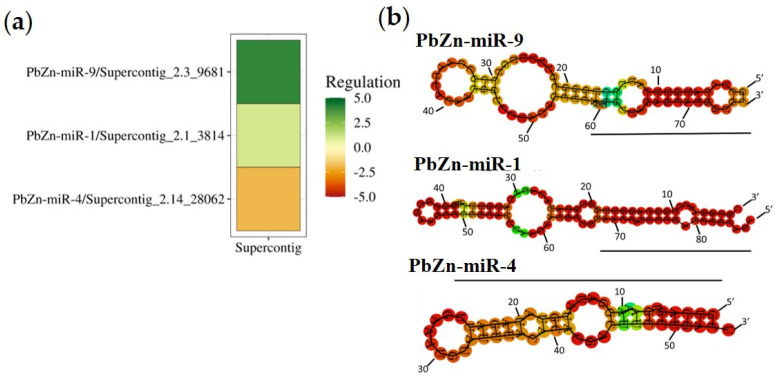
Heat map and image of differentially expressed microRNAs during zinc deprivation. (**a**) Heat map: The significant miRNAs observed are plotted with a fold change value in each color-coded histogram, with the grades of green for up-regulated genes and grades of red for down-regulated genes. (**b**) Structural representation of differentially expressed microRNAs.

**Figure 2 jof-09-00281-f002:**
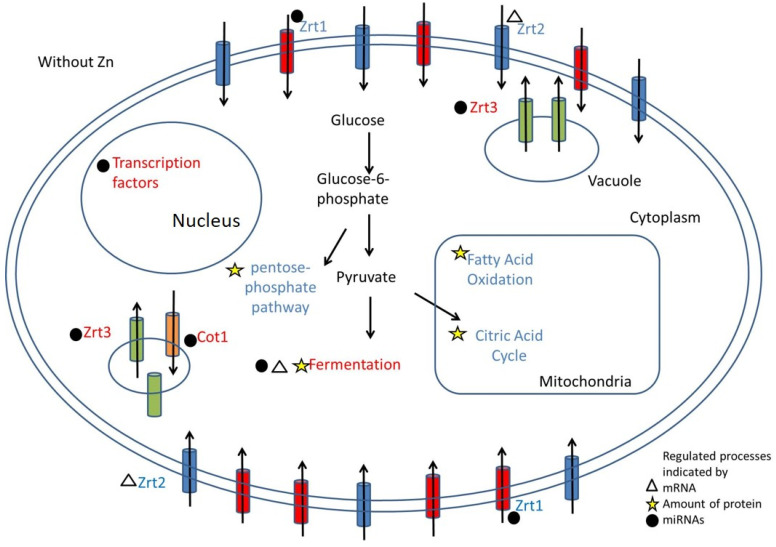
Hypothetical model of the response to zinc deprivation of *P. brasiliensis*. Letters in blue show processes that were induced and letters in red show processes that were suppressed. The image shows the altered metabolic processes during zinc deprivation in addition to the approach used in the study that indicated this alteration. Triangles mark processes indicated by changes in mRNA. Stars show processes indicated by the proteome and spheres indicate changes indicated by miRNAS.

**Table 1 jof-09-00281-t001:** MiRNAs present in *P. brasiliensis* yeast cells, cultured in the presence or deprivation of zinc.

Library ^a^	Mature Sequence	Star Sequence	Precursor Sequence	Id ^b^
C1, C2, C3, D1, D2, D3	uaaccaugucgaucugcaga	cgcggaacccggcagguuggu	cgcggaacccggcagguuggugcauauauauauguucauuggcgggaagccgggcauuuaagaaugcuguaaccaugucgaucugcaga	PbZn-miR-1 *
C1, C3, D1, D2, D3	uuauuuuuggaacuuuuu	uaguucggagucugggu	uaguucggagucuggguauggcuuucuuucuguaaugguuuuguuauguuauuuuauuuuuggaacuuuu	PbZn-miR-2 *
C1, C2, C3, D1, D2, D3	uauaucaggguguguguuggc	caaccccugauugguagg	uauaucaggguguguguuggcagcaagauauugcaaagagccaaccccugauugguagg	PbZn-miR-3 *
C1, C2, C3, D1, D2, D3	gucagugaaaggauauguauagauu	uuacacacgacuuccugacu	gucagugaaaggauauguauagauugaaauguauuuacacacgacuuccugacu	PbZn-miR-4 *
C1, C2, C3, D1, D2, D3	uucuuuaugaacaguggcugg	agucauucauugagaauu	agucauucauugagaauuucucaagcagguucuuuaugaacaguggcugg	PbZn-miR-5 *
C1, C2, C3, D1, D2, D3	ugccuuguagauaucuaaga	ucuuuucaguaggacaca	ucuuuucaguaggacacaucuauauuagaagugccuuguagauaucuaaga	PbZn-miR-6
C1, C3, D1, D2, D3	uaggucugagacagcuca	agcuguucagaccagu	uaggucugagacagcucacucagagagggcuucaaaaugcuaagaguggacaaucuucucuuauauagcuguucagaccagu	PbZn-miR-7
C1, C2, C3, D1, D2, D3	cuaguuagguuaguuaguua	gcuaacuaacuaacuaaau	gcuaacuaacuaacuaaauaacuaacuaacuaaccuaguuacuuaacuaguuagguuaguuaguua	PbZn-miR-8 *
C1, C2, C3, D1, D2, D3	aguugguuagagcauggugc	gcuacauguucauuccugg	gcuacauguucauuccugggcguuuuuguccgccaaucacaacgguccuauagcucaguugguuagagcauggugc	PbZn-miR-9
C1, C3, D1, D2	cacccguggacugugccaugc	auuguacggagcacggguguc	cacccguggacugugccaugcaucuacgugcauuguacggagcacggguguc	PbZn-miR-10
C1, C2, C3, D1, D2, D3	uaguuagguuaguuaguua	acuaaguuaguuaacaguuaac	acuaaguuaguuaacaguuaacuaacuaguuauuuaguuaacuaguuaguuaguuagguuaguuaguua	PbZn-miR-11 *
C1, C2, D1, D2, D3	aucuugacugucgaaaggg	cuagacagaucaugugaucu	cuagacagaucaugugaucuccauauccgcuggggaucuugacugucgaaaggg	PbZn-miR-12
C1, C2, C3, D1, D2, D3	ucagaugaugaaaaagaugcugaca	ucagacuggucucugcugaau	ucagacuggucucugcugaaucuccacugcggcaaauggaaguuucagaugaugaaaaagaugcugaca	PbZn-miR-13 *
C1, C2, C3, D1, D2, D3	guggaugaugagagaacuucugagg	uuagcaagugccauuauccauga	guggaugaugagagaacuucugagggcugaacaggagguccuuuagcaagugccauuauccauga	PbZn-miR-14
C1, C2, C3, D1, D2, D3	aaugggcacuguuaacuaacuu	guugguuaacgguugguuacuaa	uugguuaacgguugguuacuaagaaugggcacuguuaacuaacuu	PbZn-miR-15
C1, C3, D1, D2	uacuuuuucgauugaggggacgu	gucguucuuaaucccgcg	uacuuuuucgauugaggggacguuccgaggaggagcgucguucuuaaucccgcg	PbZn-miR-16
C1, C2, C3, D1, D2, D3	ucggaagaugaugaacgagcgg	guucaacugcaucaucaucacc	guucaacugcaucaucaucacccccucuagaggcagggucggaagaugaugaacgagcgg	PbZn-miR-17 **
C1, C2, C3, D1, D2, D3	uaagacgcgaacuguuugaggu	uuauccaacgguucccauuugg	uaagacgcgaacuguuugagguuucguagaauuauccaacgguucccauuugg	PbZn-miR-18 *
C1, C3, D1, D2, D3	uggaguggucgcccaggcug	gcccaacacccccacgcgcc	uggaguggucgcccaggcugcuagggcccaacaacccaauuagggcccaacagcccaacacccccacgcgcc	PbZn-miR-19

^a^ C-Control, D-zinc deprivation. * MicroRNAs identified in previous work (de Curcio et al., 2019) [60]. ** MicroRNAs identified in previous work (de Curcio et al., 2021) [42]. ^b^ Identification of miRNAs.

**Table 2 jof-09-00281-t002:** Targets of differentially expressed miRNAs confirmed by transcriptional or proteomic analysis.

MicroRNA	Superconting	Targets	MFE	Functional Classification	*p*-Value		Confirmed by
PbZn-miR-1	Supercontig_2.1_3814	PADG_03627/2-oxoisovalerate dehydrogenase subunit beta	−23.6 kcal/mol	Amino acid metabolism	0.033793	3′UTR	Proteome
PbZn-miR-1	Supercontig_2.1_3814	PADG_02134/Coatomer subunit epsilon	−24.1 kcal/mol	Cellular transport, transport facilities and transport routes	0.027341	3′UTR	Proteome
PbZn-miR-1	Supercontig_2.1_3814	PADG_05239/Tubulin-specific chaperone A	−22.9 kcal/mol	Cytoskeleton/structural proteins	0.045399	3′UTR	Proteome
PbZn-miR-9	Supercontig_2.3_9681	PADG_08119/Fumarate hydratase, mitochondrial	−23.5 kcal/mol	Tricarboxylic-acid pathway (citrate cycle, Krebs cycle, TCA cycle)	0.035252	3′UTR	Proteome
PbZn-miR-9	Supercontig_2.3_9681	PADG_03403/aldehyde dehydrogenase	−25.0 kcal/mol	Cell rescue, defense and virulence	0.018564	3′UTR	Proteome
PbZn-miR-9	Supercontig_2.3_9681	PADG_08409/hypothetical protein	−24.2 kcal/mol	Unclassified proteins	0.026205	3′UTR	Proteome
PbZn-miR-9	Supercontig_2.3_9681	PADG_01174/alcohol dehydrogenase	−23.3 kcal/mol	Fermentation	0.038359	5′UTR	Transcriptome
PbZn-miR-9	Supercontig_2.3_9681	PADG_04701/alcohol dehydrogenase	−25.4 kcal/mol	Fermentation	0.015719	5′UTR	Proteome
PbZn-miR-9	Supercontig_2.3_9681	PADG_05798/ssDNA binding protein, putative	−22.9 kcal/mol	Cell cycle and DNA processing	0.045208	5′UTR	Proteome
PbZn-miR-9	Supercontig_2.3_9681	PADG_04966/phosducin family protein	−23.8 kcal/mol	Protein fate (folding, modification, destination)	0.031050	5′UTR	Proteome
PbZn-miR-9	Supercontig_2.3_9681	PADG_03258/lipid A export ATP-binding/permease protein msbA	−25.6 kcal/mol	Cellular transport, transport facilities and transport routes	0.014432	5′UTR	Transcriptome
PbZn-miR-9	Supercontig_2.3_9681	PADG_03130/Myosin-10	−23.8 kcal/mol	Biogenesis of cellular components	0.031050	5′UTR	Transcriptome
PbZn-miR-9	Supercontig_2.3_9681	PADG_05949/ hypothetical protein	−23.2 kcal/mol	Unclassified proteins	0.040012	5′UTR	Transcriptome
PbZn-miR-4	Supercontig_2.14_28062	PADG_00171/cytochrome b2	−24.6 kcal/mol	Electron transport and membrane-associated energy conservation	0.029065	3′UTR	Proteome
PbZn-miR-4	Supercontig_2.14_28062	PADG_00806/BTB domain-containing protein	−23.6 kcal/mol	Translation	0.043859	5′UTR	Transcriptome
PbZn-miR-4	Supercontig_2.14_28062	PADG_07281/Guanine nucleotide-binding protein subunit alpha	−23.5 kcal/mol	Cellular communication/signal transduction mechanism	0.045692	5′UTR	Proteome

## Data Availability

The small RNA sequences were submitted to the Short Read Archive of the NCBI under BioProject number PRJNA931606 and are available here: https://www.ncbi.nlm.nih.gov/bioproject/PRJNA931606 (accessed on 5 February 2023). The proteome was deposited in the ProteomeXchange via the PRIDE database with the accession number: PXD039807. The mRNA sequences were submitted to the NCBI under BioProject number PRJNA931668 and are available here: https://www.ncbi.nlm.nih.gov/bioproject/PRJNA931668 (accessed on 5 February 2023).

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
