# Peer review of "Global Molecular Response of Paracoccidioides brasiliensis to Zinc Deprivation: Analyses at Transcript, Protein and MicroRNA Levels"

_jof, 2023, doi:10.3390/jof9030281_

Round 1

Reviewer 1 Report

The authors described the Global molecular response of Paracoccidioides brasiliensis to zinc deprivation. The manuscript is scientifically informative on basic molecular biological fields of pathogenic fungi. The experimental approaches used by the authors are also appropriate and data provided are in good quality. In my opinion, this manuscript is worth to publish, but still I believe the manuscript is needed to revise and acceptable after major revisions for the publication in “Journal of Fungi”.

Major;

- There are numerous metal chelators such as TPEN and BAPTA. Why the authors used DTPA as zinc deprivation? Please describe the reason.

- The authors should be confirm transcriptomics (RNAseq) data by qRT-PCR.

Minor;

- Title; Paracoccidioides to Italic

- Line 140; BHI to full name.

Author Response

Please, see below the responses to reviewer 1:

According to she/he:

1-There are numerous metal chelators such as TPEN and BAPTA. Why the authors used DTPA as zinc deprivation? Please describe the reason.

Response: DTPA (diethylenetriaminepentaacetic acid), TPEN [N,N,N’,N’-tetrakis(2-pyridylmethyl) ethylenediamine] and BAPTA [1,2-bis(o-aminophenoxy)ethane-N,N,N’,N’-tetraacetic acid are usually used for metal ion chelation. BAPTA is used as a calcium chelator (Alberdi et al., 2001; Stork and Li, 2006)  [1,2]. TPEN is an intracellular membrane-permeable ion chelator, that chelates not only the extracellular zinc ions but also those inside the cell. On the other hand, DTPA is a membrane-impermeable zinc chelator, that accesses zinc only in the extracellular environment. Zinc depletion induced by TPEN induced apoptosis in hepatocytes, while the DTPA did not (Nakatani et al., 2000) [3]. Therefore, TPEN is a stronger chelator and may affect cell metabolism at a toxic level. For this reason, DTPA was our choice.

2- The authors should be confirm transcriptomics (RNAseq) data by qRT-PCR.

Response: Unfortunately, when we went to perform the validation of some of our DEG with qPCR our RNA samples were highly degraded.

However, we dispute the need to perform a qPCR validation on RNA-Seq data. The RNA-Seq technique is very robust and highly precise. The need to validate gene expression data with qPCR comes from the time when people were using microarray for high-throughput gene expression essays. Microarrays have many more sources of biases, like the effect of probe within gene, that can affect the results.

There have been many discussions among scientists about the usefulness of qPCR to validate RNA-Seq results. Please, take a look at some of these discussions:

https://twitter.com/jrossibarra/status/785493380549660672

https://rna-seqblog.com/do-i-need-to-validate-my-rna-seq-results-with-qpcr/

http://bridgeslab.sph.umich.edu/posts/validation-of-rnaseq-experiments-by-qpcr

https://www.researchgate.net/post/Why_is_qPCR_required_to_validate_RNAseq_result

https://www.researchgate.net/post/Is_it_necessary_to_do_qPCR_to_validate_genes_from_RNA-sequence

As you can see in these discussions, many scientists believe qPCR is not any more accurate than RNA-Seq. RNA-Seq has be proven to be a very accurate and robust technique. Therefore, there is no need to technically validate its results. We argue for the importance of further functional investigation, like we did with proteomics.

http://bridgeslab.sph.umich.edu/posts/validation-of-rnaseq-experiments-by-qpcr

https://www.researchgate.net/post/Why_is_qPCR_required_to_validate_RNAseq_result

https://www.researchgate.net/post/Is_it_necessary_to_do_qPCR_to_validate_genes_from_RNA-sequence       

Minor;

  • Title; Paracoccidioides to Italic
  • BHI: brain heart infusion
  1. Alberdi, A.; Jimenez-Ortiz, V.; Sosa, M.A. The Calcium Chelator BAPTA Affects the Binding of Assembly Protein AP-2 to Membranes. Biocell 2001, 25, 167–172.
  2. Stork, C.J.; Li, Y. v. Intracellular Zinc Elevation Measured with a “Calcium-Specific” Indicator during Ischemia and Reperfusion in Rat Hippocampus: A Question on Calcium Overload. Journal of Neuroscience 2006, 26, 10430–10437, doi:10.1523/JNEUROSCI.1588-06.2006.
  3. Nakatani, T.; Tawaramoto, M.; Opare Kennedy, D.; Kojima, A.; Matsui-Yuasa, I. Apoptosis Induced by Chelation of Intracellular Zinc Is Associated with Depletion of Cellular Reduced Glutathione Level in Rat Hepatocytes. Chem Biol Interact 2000, 125, 151–163, doi:10.1016/S0009-2797(99)00166-0.

Reviewer 2 Report

Major comments:

 - Raw data needs to be deposited in a repository.

- Section 2.9 Why was DESeq2 used for microRNAs but edgeR for mRNA-seq. Would be more consistent to use one? Especially if comparing the datasets, this needs to be addressed. If integrating multiple datasets but methods are different for no reason, it can't be done.

- L70 This reference is not formatted right.

- L140 BHI medium needs a ref or description

- L151 The trizol methods needs description or a ref.

- L153 "The extracted RNA... microRNAs" describe the library prep. What kit was used, assuming there was some sort of rRNA depletion. How was this done specifically for P. brasiliensis. Assume it was not polyA enrichment because that wouldn't capture all miRNAs.

- L161: Bolger et al needs formatting.

- L164: What version and genome was used for running the program?

- L217: how was cDNA made, what kit?

- Table 1 is too big, needs to be condensed in order to be read.

Author Response

According to reviewer 2:

1-Raw data needs to be deposited in a repository.

The proteome was deposited in the  ProteomeXchange via the PRIDE database with the accession number: PXD039807. This information has been added.

The small RNA sequences were submitted to the Short Read Archive of the NCBI under BioProject number PRJNA931606 and are available here: https://www.ncbi.nlm.nih.gov/bioproject/PRJNA931606.

The mRNA sequences were submitted to the NCBI under BioProject number PRJNA931668 and are available here: https://www.ncbi.nlm.nih.gov/bioproject/ PRJNA931668.

2- Section 2.9 Why was DESeq2 used for microRNAs but edgeR for mRNA-seq. Would be more consistent to use one? Especially if comparing the datasets, this needs to be addressed. If integrating multiple datasets but methods are different for no reason, it can't be done.

Reviewer is correct, DESeq2 and edgeR were used for differential expression of microRNA and mRNA-Seq, respectively. To be sincere, this inconsistency arose because two different bioinformaticians were involved in these analyses. However, we performed the microRNA differential expression analysis again with edgeR. The same three microRNAs, identified with DESeq2, were differentially expressed in the results of edgeR. Therefore, we changed the material and methods declaring that both differential expression analysis was performed with edgeR.

3-- L70 This reference is not formatted right.

Response: ok

- L140 BHI medium needs a ref or description

Response: This information has been added.

Brain Heart Infusion medium

4- L151 The trizol methods needs description or a ref.

This information has been added.

de Curcio et al., 2021 [1]

5- L153 "The extracted RNA... microRNAs" describe the library prep? What kit was used, assuming there was some sort of rRNA depletion. How was this done specifically for P. brasiliensis. Assume it was not polyA enrichment because that wouldn't capture all miRNAs.

The kit used was, NEBNext® Multiplex Small RNA Library Prep Set for Illumina (Illumina KIT).  More information about library staging has been included such as ligation of adapters and separation of smaller fragments in agarose gel. The material sent is just RNA obtained from the fungus under controlled cultivation conditions (we evaluated the macroscopy and microscopic characteristics of the culture). In addition as described and included in the manuscript all sequences after processing were submitted to BLAST in NCBI to confirm that they were from P. brasiliensis.  Other works published by the group used the same methodology:

de Curcio et al., 2021 [1]

de Curcio et al., 2018 [2]

This information has been added:

6-Lines 152-159: The libraries were constructed with the NEBNext® Multiplex Small RNA Library Prep Set for Illumina (Illumina KIT). Initially, adapters were ligated to the 3' and 5'regions of the RNA molecules, then cDNAs were synthesized, amplified with SuperScript II Reverse Transcriptase with 100 mM DTT and 5X First Strand Buffer (Invitrogen) purified and the size selected from an agarose gel. The sequencing of the samples was performed by GenOne Biotechnologies (www.GenOne.com.br) Rio de Janeiro-RJ-Brazil, using the Illumina HiSeq 2500 platform

7-Lines 166-169 : After processing the sequences, they were submitted to BLASTx [3] against the nr database (https://blast.ncbi.nlm.nih.gov/) in order to confirm that the sequences of the RNAseq were from P. brasiliensis.

8-- L161: Bolger et al needs formatting.

Response: ok

9-- L164: What version and genome was used for running the program?

The genome utilized was the Paracoccidioides brasiliensis Pb18 v.2 (assembly Paracocci_br_Pb18_V2) available on NCBI (https://www.ncbi.nlm.nih.gov/genome/334). This same phrase was added in the methods section. We thank the reviewer for pointing that out.

10- L217: how was cDNA made, what kit?

This information has been added.

11-Lines 152-159: The libraries were constructed with the NEBNext® Multiplex Small RNA Library Prep Set for Illumina (Illumina KIT). Initially, adapters were ligated to the 3' and 5'regions of the RNA molecules, then cDNAs were synthesized, amplified with SuperScript II Reverse Transcriptase with 100 mM DTT and 5X First Strand Buffer (Invitrogen) purified and the size selected from an agarose gel. The sequencing of the samples was performed by GenOne Biotechnologies (www.GenOne.com.br) Rio de Janeiro-RJ-Brazil, using the Illumina HiSeq 2500 platform.

12- Table 1 is too big, needs to be condensed in order to be read.

Table 1 was the only one included in the manuscript text; all others were very big and for that were supplementary. This fact reflects the data complexity. 

References:

  1. de Curcio, J.S.; Oliveira, L.N.; Batista, M.P.; Novaes, E.; de Almeida Soares, C.M. MiRNAs Regulate Iron Homeostasis in Paracoccidioides Brasiliensis. Microbes Infect 2021, 23, 104772, doi:10.1016/J.MICINF.2020.10.008.
  2. de Curcio, J.S.; Paccez, J.D.; Novaes, E.; Brock, M.; de Almeida Soares, C.M. Cell Wall Synthesis, Development of Hyphae and Metabolic Pathways Are Processes Potentially Regulated by MicroRNAs Produced between the Morphological Stages of Paracoccidioides Brasiliensis. Front Microbiol 2018, 9, 3057, doi:10.3389/FMICB.2018.03057/BIBTEX.
  3. Altschul, S.F.; Gish, W.; Miller, W.; Myers, E.W.; Lipman, D.J. Basic Local Alignment Search Tool. J Mol Biol 1990, 215, 403–410, doi:10.1016/S0022-2836(05)80360-2.

Round 2

Reviewer 1 Report

Now, the manuscript is acceptable for publication.

Reviewer 2 Report

The authors have addressed all my comments and queries. I would like to commend them on an excellent piece of work.